# Dipeptidyl Peptidase 4 (DPP4) as A Novel Adipokine: Role in Metabolism and Fat Homeostasis

**DOI:** 10.3390/biomedicines10092306

**Published:** 2022-09-16

**Authors:** Ilaria Barchetta, Flavia Agata Cimini, Sara Dule, Maria Gisella Cavallo

**Affiliations:** Department of Experimental Medicine, Sapienza University of Rome, Viale Regina Elena 324, 00161 Rome, Italy

**Keywords:** dipeptidyl peptidase 4, adipose tissue, fat, metabolism, obesity, diabetes mellitus, NAFLD, metabolic syndrome, incretins, adipokine

## Abstract

Dipeptidyl peptidase 4 (DPP4) is a molecule implicated in the regulation of metabolic homeostasis and inflammatory processes, and it exerts its main action through its enzymatic activity. DPP4 represents the enzyme most involved in the catabolism of incretin hormones; thus, its activity impacts appetite, energy balance, and the fine regulation of glucose homeostasis. Indeed, DPP4 inhibitors represent a class of antidiabetic agents widely used for the treatment of Type 2 diabetes mellitus (T2DM). DPP4 also acts as an adipokine and is mainly secreted by the adipose tissue, mostly from mature adipocytes of the visceral compartment, where it exerts autocrine and paracrine activities. DPP4 can disrupt insulin signaling within the adipocyte and in other target cells and tissues, where it also favors the development of a proinflammatory environment. This is likely at the basis of the presence of elevated circulating DPP4 levels in several metabolic diseases. In this review, we summarize the most recent evidence of the role of the DPP4 as an adipokine-regulating glucose/insulin metabolism and fat homeostasis, with a particular focus on clinical outcomes associated with its increased secretion in the presence of adipose tissue accumulation and dysfunction.

## 1. Introduction

### 1.1. DPP4: Forms and Functions

Dipeptidyl peptidase 4 (DPP4) is a ubiquitously expressed glycoprotein which displays an exopeptidase activity [1]. DPP4 was first discovered in 1966; later on, its structure was found to be identical to the one of the Cluster of Differentiation 26 (CD26), known at that time as an activation marker of T lymphocytes [2].

The gene encoding for DPP4 is located on the long arm of the chromosome 2 (2q24.3) [3] and is preceded by a promoter region controlled by transcription factors such as the nuclear factor kappa-light-chain enhancer of activated B cells (NF-κB) [4], the hepatocyte nuclear factor 1β (HFN1β) [5], the lysine acetyltransferase 5 (KAT5) [6], and the signal transducers and activators of transcription 1α (STAT1α) [7], all centrally involved in cell growth and proliferation, immune responses, and metabolism.

DPP4 belongs to the serine peptidase/prolyl oligopeptides family, which also includes the DPP6, DPP8, DPP9, DPP10, the fibroblast activation protein (FAP), and the prolyl endopeptidase [8]. Furthermore, a number of enzymes exhibit DPP4-like activity or share a similar structure to DPP4, so these are commonly known as DPP4 activity and/or structure homologs (DASH) [9].

DPP4 is detectable in two different forms, including the transmembrane enzyme and the soluble protein, indicated as *sDPP4*, which derives from the cleavage and shedding of the transmembrane DPP4.

The transmembrane DPP4 is constituted in sequence by an extracellular C-terminal catalytic domain, a cysteine-rich region, a glycosylated area, and finally by the transmembrane segment and six cytosolic amino acids at the N-terminal [10].

The soluble form of the DPP4, instead, lacks the intracellular and transmembrane domains and is measurable in body fluids such as serum, saliva, seminal fluid, bile, and cerebrospinal liquid [11]. The bioactive forms of the sDPP4 are the homodimer and the oligomer, whereas the monomeric sDPP4 has very low residual enzyme activity [12], since it undergoes post-translational modifications, i.e., glycosylation and sialylation, that reduce its stability and catalytic activity [13,14]. With aging, the sDPP4 seems to be more sialylated and, therefore, its bioactivity is reduced [15].

The pathophysiological mechanisms behind transmembrane DPP4 shedding and sDPP4 release in the bloodstream have not been fully clarified yet. Initial evidence shows that processes leading to sDPP4 secretion are organ- and tissue-specific and mainly mediated by metalloproteases.

DPP4 exerts many biological functions through its C-terminal catalytic domain, but also by noncatalytic mechanisms.

As for its catalytic activity, DPP4 acts as an exopeptidase that cleaves peptides after the second position from the N-terminal, with preference for proline residues [16].

DPP4 deeply influences different metabolic pathways and cell processes in relation to the target peptides, exerting major influences on glucose metabolism, signal transduction, cell migration, and maturation and immune responses.

A large number of DPP4 substrates have been identified in vitro, whereas only some were studied in vivo. Among them, well-established DPP4 target peptides are Glucagon-like peptide 1 (GLP-1), Glucagon-like peptide 2 (GLP-2), Gastric Inhibitory Polypeptide (GIP), Insulin-like Growth Factor 1 (IGF-1), Neuropeptide Y, and Peptide YY, as described in Section 2.1.

Among the most relevant noncatalytic functions, DPP4 binds the adenosine deaminase (ADA) through its extracellular domain providing co-stimulatory signals to T cells [17]. Since the ADA is an enzyme involved in autoimmune diseases and cancer development [18,19], it is plausible to hypothesize that the DPP4 involvement in ADA-mediated pathways is at the basis of the elevated sDPP4 levels reported in these diseases [20,21].

Another nonenzymatic activity performed by the sDPP4 is the extracellular matrix remodeling, which is exerted by the linkage with Collagen I, Collagen III, and fibronectin through the DPP4 C-terminal noncatalytic segment [22,23].

Due to the breadth of actions carried out by DPP4 substrates, impaired sDPP4 levels and DPP4 activity have been reported in several dysmetabolic conditions, such as Type 2 diabetes mellitus (T2DM), obesity [24], metabolic syndrome, nonalcoholic fatty liver disease (NAFLD) [25], and polycystic ovary syndrome [26]. Moreover, increased DPP4 levels were also reported in cancer [20], autoimmune diseases [21], and neurodegenerative conditions [27].

### 1.2. DPP4 Expression and Secretion 

In humans, what represents the main source of circulating sDPP4 is still under debate, even if increasing data point to the adipose tissue (AT) as a primary contributor, mostly in the presence of fat mass expansion and obesity [28].

In adipocytes, DPP4 cleavage from cell membrane is mediated by a specific subset of proteases, different from those involved in all the other cell types [29], and is upregulated in response to inflammatory stimuli and elevated insulin concentrations [28].

Conversely, the major trigger for sDPP4 release from human smooth muscle cells is the hypoxia. In addition, after stimulation, no significant change of tissue DPP4 mRNA expression was reported, suggesting that greater sDPP4 levels in this condition are attributable to augmented cleavage rate rather than to increased tissue DPP4 levels [28,29].

As described in relation to circulating sDPP4 concentration, local DPP4 levels are also modulated in a tissue-specific manner. DPP4 expression has been detected in several tissues and organs, such as lung, small intestine, spleen, liver, kidney, pancreas [30], endothelial cells, hematopoietic cells, and AT [31]. Although hematopoietic and endothelial cells are substantial contributors to the overall circulating sDPP4 concentration [32], the sDPP4 derived from these cells does not seem to significantly influence inflammatory or metabolic pathways [31].

In the last decade, a large body of experimental evidence has shown that a great amount of sDPP4 is released from the adipocytes, especially from those sited in the visceral adipose tissue (VAT), and that the AT-derived sDPP4 displays unfavorable effects on insulin signaling and sensitivity. Indeed, DPP4 was finally proposed as a novel adipokine [28].

Since then, accumulating observations have reported the influence of DPP4 in metabolic pathways and the involvement of this molecule in the development of insulin-resistance-related disorders. A bulk of studies have gathered so-far growing evidence on the role of DPP4 as an adipokine, and they are discussed in the present review, which specifically focuses on the role of DPP4 in metabolic regulation and AT homeostasis. Finally, this article reviews evidence available on the relationship between DPP4 and the presence and development of the metabolic complications of body fat expansion and obesity.

## 2. DPP4 and Metabolism: Role of DPP4 in the Regulation of Glucose Metabolism

Many of the substrates cleaved by the DPP4 belong to the class of the incretin hormones, such as the GLP-1, GLP-2, and GIP, and this explains the importance of this molecule in the regulation of glucose–insulin metabolism. Specific inhibitors of the DPP4 (DPP4-Is) have been developed and approved as a therapeutic strategy for T2DM treatment [33] and, more recently, they have been proposed as a first-step therapy in individuals with latent autoimmune diabetes of the adults (LADA) [34], for the broad metabolic pathways mediated by the DPP4/incretin system [35].

Many data produced in animal models demonstrated the presence of a major contribution of DPP4 in the regulation of glucose metabolism: DPP4-deficient rats had better glucose tolerance, higher insulin, and GLP-1 levels than DPP4-positive rats, and glucose tolerance improved when wild-type rats were treated with DPP4-Is [36,37,38,39,40].

The central contribution of GLP-1 to DPP4 effects on glucose metabolism was not confirmed by other reports, which showed that treatment with DPP4-Is also improved glucose tolerance in GLP-1-deficient rats [37].

It is extremely likely that the list of DPP4 substrates identified so far is inexhaustive. Some substrates have been studied both in vitro and in animal models; however, confirming the DPP4 cleaving activity in the clinical setting is of the utmost challenge, due to the very low serum concentrations of some peptides and the unavailability of sufficiently sensitive methods for their detection [41,42,43].

In the following paragraphs we discuss different DPP4 mechanisms that impact glucose metabolism, starting from its role as a cleaving peptide on ascertained substrates that have a recognized role in glucose/insulin homeostasis in humans.

### 2.1. DPP4 and Its Substrates: Incretin System

#### 2.1.1. Glucagon-like Peptide 1 (GLP-1)

The enteroendocrine intestinal cells secrete incretin hormones that promote insulin release. The GLP-1 and GIP belong to the incretin hormone family and are responsible for almost 70% of the insulin secreted after a meal [44]. Specifically, the intestinal L cells express the proglucagon gene, which contains regions encoding for glucagon, GLP-1, GLP-2, oxyntomodulin (OXM), glicentin, and glicentin-related pancreatic peptide [45].

The biologically active forms of GLP-1 are the fragments GLP-1^(7−37)^ and GLP1^(7−36)^, cleaved by the DPP4 to generate inactive GLP-1^(9−37)^ and GLP-1^(9−36)^ [46].

The GLP-1 acts by promoting insulin release and inhibiting glucagon secretion after food ingestion; it delays stomach emptying and inhibits appetite [47]; these mechanisms were shown to be impaired in patients with T2DM who have less differentiated enteroendocrine cells and, as a consequence, impaired GLP-1 response to food ingestion [48].

#### 2.1.2. Glucagon-like Peptide 2 (GLP-2)

As described for the GLP-1, the active GLP-2^(1−33)^ is also released postprandially by the intestinal L cells and cleaved by DPP4 to its inactive GLP-2^(3−33)^ form.

The biologically active GLP-2 decreases gut motility, gastric acid secretion [45], and has an intestinotrophic action, since it reduces enterocyte apoptosis, increases crypt cell proliferation, and improves intestinal blood flow [49]. For these properties, it has been studied as a therapeutic target for the treatment of short bowel syndrome and Crohn’s disease [49].

GLP-2 receptors have been identified in central nervous system cells, including proopiomelanocortin (POMC) neurons, suggesting a role of this incretin in the central control of food intake, body weight, and glucose metabolism [50].

Experiments in mice showed that treatment with a GLP-2 analogue reduced VAT inflammation and improved insulin sensitivity, without decreasing total fat mass or body weight. When the GLP-2 receptor is specifically silenced at the VAT level, all the GLP-2 beneficial effects on glucose metabolism and inflammation are lost [51].

This makes plausible the hypothesis that GLP-2 may display beneficial effects on metabolism by ameliorating the VAT micro-environment in relation to its anti-inflammatory properties. In line with this observation, DPP4 may worsen the metabolic homeostasis in the presence of fat accumulation by disrupting the GLP-2 mediated regulation of VAT inflammatory balance (see also Section 4.2).

#### 2.1.3. Glucose-Dependent Insulinotropic Peptide (GIP)

GIP is another major DPP4 substrate, it is secreted by the enteroendocrine K cells of the upper intestinal tract in a nutrient dependent manner, and it promotes insulin and glucagon secretion [44]. GIP is quantitatively the most important incretin hormone able to stimulate the insulin secretion, unlike what is observed for the GLP-1, which regulates glucose concentration mainly by inhibiting glucagon release [44]. The biologically active GIP^(1−42)^ is degraded by DPP4 into inactive GIP^(3−42)^ [46]. It increases the gastric emptying rate, inhibits gastric acid secretion, and reduces intestinal transit; the effects of GIP on satiety and food intake remain, instead, controversial [52].

#### 2.1.4. Neuropeptide Y (NPY), Peptide YY, and Pancreatic Polypeptide

The NPY, peptide YY, and pancreatic polypeptide are structurally similar peptides belonging to the NPY system and are centrally involved in the regulation of β cell biology. They bind five receptors (Y1R, Y2R, Y4R, Y5R, and Y6R), determining receptor- and tissue-specific responses: i.e., the activation of Y1R in the brain has orexigenic effects and decreases anxiety, whereas the Y2R activation leads to anxious feelings and reduces appetite [53]. NPY^(1−36)^ is cleaved by DPP4 into inactive NPY^(3−36)^, which is unable to bind Y1R [54]. Peptide YY^(1−36)^ is truncated by DPP4 into peptide YY^(3−36)^ [55].

NPY is localized mostly in the central and peripheral nervous system, but also in pancreatic β cells and bone [55]; in the pancreas, different NPY system members are expressed in different cell types, as NPY is expressed by β cells [56,57], peptide YY by α cells, and pancreatic polypeptide by PP cells [58]. NPY–Y1R binding at the pancreatic level determines inhibitory effect on insulin secretion [59].

Peptide YY is expressed in the central nervous system, where it reduces appetite, and is also released by the intestine cells along with GLP1 after food ingestion, delaying gastric emptying, reducing gut motility and pancreatic exocrine secretions [59,60].

Finally, impaired DPP4 levels may negatively affect central food intake control, brain–gut axis, and glucose homeostasis, also influencing concentrations and bioactivity of specific neuroendocrine peptides.

#### 2.1.5. Insulin-like Growth Factor 1 (IGF-1)

Insulin-like Growth Factor 1 (IGF-1) is involved in glucose metabolism though several pathways converging on insulin signaling, in relation to its insulin-like structure and function, and can bind and activate both insulin and hybrid insulin/IGF-1 receptors [61].

IGF-1 is released mainly by the liver in response to elevated insulin and growth hormone (GH) levels, such as after food ingestion, when insulin levels increase, leading to augmented free IGF-1 concentrations in the bloodstream. IGF-1 also promotes glucose uptake in muscle cells; in humans, IGF-1 levels are reduced in condition of insulin resistance, and low IGF-1 levels are associated with the presence of diabetes’ complications [61].

The IGF-1 linkage with its specific receptor IGF-1R activates downstream pathways involving the Phosphatidyl Inositol 3-Kinase (PI3K), the protein kinase B (Akt), and the Extracellular signal-Regulated Kinase 1/2 (ERK1/2), finally leading to stimulate cell proliferation and differentiation [61,62]. The bioactive IGF-1^(1−70)^ is a substrate of DPP4; when truncated to IGF-1^(3−70)^, it decreases its affinity for IGF-1R [62].

#### 2.1.6. Stromal-Derived Factor 1 (SDF-1)

The SDF-1 is a widely expressed chemokine, released in response to cell damage, with the purpose of stimulating the migration of hematopoietic and endothelial progenitor cells to the injury site [52]. For these properties, it is involved in myocardial protection, bone repair, and neuronal regeneration, but also in metastasis formation [38]; both the active isoforms of SDF-1 are truncated by DPP4 into inactive peptides [39].

As for metabolism regulation, SDF-1 expression is upregulated in conditions of insulin resistance in animal models, human adipocytes, and in obese individuals with T2DM [39]; SDF-1 significantly reduces glucose uptake by adipocytes IRS-1 levels, attenuating Akt phosphorylation, and impairing insulin signaling [39,63].

#### 2.1.7. Substance P

Substance P is a neurotransmitter involved in the modulation of emotional behavior, pain transmission, and neuroinflammation [64]; in nonneuronal tissues, it mediates immune responses and cardiovascular physiopathology [64]. The active substance P^(1−11)^ is truncated in substance P^(3−11)^ and P^(5−11)^ by DPP4 and other enzymes, such as the angiotensin converting enzyme and the neutral endopeptidase [65,66].

Growing evidence demonstrates a role of this neurotransmitter in glucose metabolism. In mice with diet-induced obesity, Ramalho and collaborators showed that treatment with an antagonist of substance P-receptor (NK1-R) resulted in lower body weight gain, reduced food intake, smaller adipocytes, and lower glycaemia and insulinemia compared to the placebo group [67]. Substance P enhances lipid accumulation into adipocytes and increases local Glucose Transporter type 4 (GLUT-4) expression in presence of high glucose concentrations [68]. Mouse models with T2DM treated with substance P showed a delay in the development of glucose metabolism impairment and were protected, to some extent, from β cell damage [69].

Taken together, data available so far show that the DPP4 is centrally involved in body weight control, insulin signaling, and glucose homeostasis. Many of its pathophysiological activities are exerted by regulating the half-life of hormones belonging to the incretin system that are primary regulators of food intake and energy expenditure at the central nervous system level and orchestrate metabolic processes after food ingestion.

Moreover, DPP4 plays a major role in modulating immune and inflammatory pathways. Several studies show that the DPP4 is a fine regulator of micro-environment homeostasis within tissues and organs that are centrally implicated in metabolism.

Among them, the adipose tissue represents both a primary source and target organ of DPP4, leading to important consequences on metabolic pathways in the presence of elevated DPP4 levels, as described in the following sections.

### 2.2. DPP4 and Insulin Resistance

Several studies show that sDPP4 levels are associated with the presence of insulin resistance in both animal models and humans, and experimental data point towards a role of DPP4 as a proper mediator, and not only a biomarker, of impaired insulin sensitivity [70,71,72].

Animal models of complete DPP4 silencing demonstrate that the DPP4 contributes to the development of obesity and insulin resistance in the condition of chronic caloric excess and positive energy balance [42,73]. DPP4-deficient rats fed with a high-fat diet were protected from body weight gain and insulin-resistance-related diseases [42], the total fat mass was not expanded, and they did not develop fat cells hypertrophy in white and brown AT or pancreatic islet hypertrophy after diet intervention [42]. Furthermore, the VAT was proportionally less represented in DPP4-deficient rats than control animals [73].

Lamers and collaborators provided the first demonstration that sDPP4 is released by human adipocytes and AT-associated macrophages in response to elevated insulin levels and proinflammatory environment, especially within the VAT [28].

The selective DPP4 knockout in the AT made it possible to demonstrate that the AT is the main source of circulating sDPP4 [74]; phenotypically, AT-DPP4-deficient animals were characterized by greater insulin sensitivity, assessed by hyperinsulinemic euglycemic clamp, and glucose tolerance, despite increased fat mass and body weight [74].

In vitro experiments provided important insights on mechanisms behind the DPP4 modulation of insulin signaling, showing that sDPP4 promotes insulin resistance by autocrine actions on adipocytes and by endocrine effects on skeletal and smooth muscle cells, overall decreasing Akt phosphorylation and impairing insulin signaling [28]. Treatment with DPP4-Is was capable of reversing these alterations [28,75].

On the other hand, DPP4 silencing or inhibition in human adipocytes favors the phosphorylation of Akt, and increases the levels of the insulin receptor and the insulin receptor substrate 1 (IRS-1) without upregulating their expression [76].

AT-specific DPP4 deletion positively impacts on hepatic insulin sensitivity in mice, suggesting a role of sDPP4 in the liver–AT cross-talk: the Insulin-like Growth Factor-Binding Protein 3 (IGF-BP3) was hypothesized to represent a potential mediator of improved hepatic insulin sensitivity in the presence of AT DPP4 knocking-out, as it is the main regulator of IGF-1 bioavailability [74].

### 2.3. DPP4, Emotional Behavior, and Physical Activity

Another mechanism underlying the relationship between increased DPP4 and the presence of metabolic alterations is the involvement of this adipokine in the regulation of emotional behavior and physical activity.

DPP4-deficient rats exhibit lower water and food intake, increased energy expenditure, and consequent body weight loss, as well as increased pain sensitivity, and reduced stress-like responses and anxiety, and they develop less sedative effects from ethanol [42,73,76,77].

The onset of these effects is largely attributable to the lack of cleaving activity exerted by the DPP4 on a number of peptides expressed in both the central nervous system and the gastrointestinal tract, i.e., GLP-1, NPY, and Peptide YY, and involved in these pathways [45,46,47,48,53,54,55,56,57], as discussed in the previous section.

Conversely, mice exposed to a high-fat, high-sucrose diet and treated with the DPP4-I anagliptin exhibited reduced food intake and body weight gain compared to nontreated mice, without changes in energy expenditure [78]. In these experiments, the mRNA expression levels in the hypothalamus of neuropeptides controlling feeding behavior, such as NPY, POMC, and Agouti-related peptide (AGRP), were comparable to those measured in control mice, suggesting a role of DPP4 in controlling food intake that was not dependent on these mediators [78]. 

Conversely, the anagliptin-treated rats showed enhanced leptin effects on food behavior and body weight; therefore, the interaction with leptin circuits may be at the basis of DPP4-mediated control of appetite [78]. Specific DPP4 inhibition in rodents’ hindbrain reduced body weight and spontaneous food intake, especially the appetite for high-fat food [79].

However, clinical studies investigating the role of DPP4 in controlling food intake in humans have led to different outcomes from those obtained in animal models.

In T2DM subjects and healthy volunteers, treatment with the DPP4-I vildagliptin showed no effect on satiation, gastric volume, intragastric pressure, or epigastric symptoms after liquid meal, water intake, or nutrient ingestion, despite increased GLP-1 levels [80,81].

Conversely, DPP4 inhibition seems to produce beneficial effects on energy expenditure [82]. In the investigation conducted by Heruc and collaborators [82], healthy males receiving vildagliptin before intra-duodenal fat infusion exhibited increased resting energy expenditure compared with placebo-treated individuals, without changes in energy intake [82].

Randomized clinical trials conducted in individuals with T2DM undergoing DPP4-Is treatment have generally demonstrated the neutral effect of these agents on body weight, as shown in meta-analyses, including all the DPP4-Is approved for T2DM treatment [83,84]. Nonetheless, a more recent meta-analysis showed that treatment with the DPP4-I sitagliptin, in monotherapy or in combination with metformin, leads to significant weight loss if the duration of treatment is over six months [85].

### 2.4. DPP4 and Pancreatic β Cell Survival

Many experiments have suggested that DPP4 may exert detrimental effects on pancreatic β cell survival.

In DPP4-deficient rats with streptozotocin-induced diabetes, the onset of hyperglycemia was delayed and circulating insulin levels were higher than in wild types [42,86]; similarly, treatment with DPP4-Is improved glucose tolerance and insulin sensitivity in diabetic rats, despite an overall increased nutrient intake and weight gain, with ameliorated β cell morphology and functionality [86]. Obese diabetic rats treated with the DPP4-I vildagliptin also displayed reduced triglyceride concentration in the pancreatic islets [77,87].

To better understand the protective effect of DPP4-I treatment against β cell damage, Hamamoto et al. [88] investigated the gene expression profiles in pancreatic islets from rats treated with vildagliptin, observing an upregulation of genes involved in cell differentiation, proliferation, and oxidative stress, and downregulation of endoplasmic reticulum stress-related genes and genes involved in cellular apoptosis [88].

For the promising effect on β cell protection, DPP4-Is have been studied in humans for the treatment of Type 1 diabetes (T1DM) and LADA and data showed better glycemic control in patients treated with DPP4-Is in addition to a traditional insulin regimen in comparison to insulin alone [89].

Sitagliptin had no effect on β cell function in individuals with long-standing T1DM [90], but it improved β cell functionality [91,92] and insulin sensitivity in subjects with LADA [91], inducing changes of T cell phenotype and gene expression profile [93]; indeed, DPP4-I treatment is nowadays considered as a therapeutic option in LADA subjects [36].

In patients with T1DM, both vildagliptin and sitagliptin were shown to inhibit the glucagon secretion in the course of hyperglycemia, without impacting the glucagon response in the condition of hypoglycemia [94,95,96].

Finally, taken together, these data show that DPP4 may negatively affect insulin sensitivity and secretion, favoring mechanisms leading from chronic caloric excess and positive energy balance to body weight gain and development of insulin-resistance-related diseases. Furthermore, the influence exerted by the DPP4 on β cell function may plausibly accelerate the onset of T2DM in high-risk populations.

## 3. DPP4 and Lipid Metabolism

The existence of a relationship between DPP4, lipid metabolism, and circulating lipid profile has been assessed in some investigations, although, to the best of our knowledge, no study has been specifically designed to explore lipid outcomes of differential DPP4 levels.

Knocked-out mice for AT DPP4 displayed lower plasma cholesterol than wild-type mice, in the presence of comparable triglycerides levels, when exposed to obesogenic diet regimen [74]. Mutant rodents also exhibited lower degrees of insulin resistance and, specifically, had better hepatic insulin sensitivity and glucose tolerance than wild types.

Indeed, the specific effect of AT DPP4 deletion on blood cholesterol could not be confirmed by this observation, and may plausibly be attributable to overall more favorable metabolic profile in absence of DPP4 at the AT level [74].

Conversely, in vitro investigations exploring whether the DPP4 might play a direct role in the adipogenesis demonstrated that murine pre-adipocytes treated with recombinant DPP4 had more cytoplasmatic lipid droplets and increased peroxisome proliferator-activated receptor gamma (PPAR-γ) expression than nontreated cells; these effects were mediated by the DPP4-induced cleavage of active NPY into NPY^(3−36)^, which induces adipogenesis [97].

An association between sDPP4 and blood triglyceride levels was observed in humans [71]; in line with clinical observations, DPP4-deficient rats exhibited reduced triglycerides, along with lower blood transaminases and improved glucose tolerance, compared to DPP4-positive animals [98].

Again, the specific contribution of the DPP4 deletion to the improved blood lipid profile in these models could not be established, since it may reflect the overall amelioration of the metabolic homeostasis in mutant rodents. In addition, treatment with the DPP4-I vildagliptin did reduce serum triglycerides in obese diabetic rats, along with triglycerides content into the pancreatic islets, with favorable effects on β cell function and insulin secretion [88].

In conclusion, some investigations, mostly performed in animal models, showed the existence of a relationship between greater DPP4 and unfavorable lipid profile.

However, DPP4 deficiency/inhibition was associated with an overall improvement of metabolic profile, in terms of insulin sensitivity and glucose tolerance, leading to inconclusive evidence on the specific contribution of DPP4 to lipid metabolism.

## 4. DPP4 in Adipose Tissue and Fat Distribution

### 4.1. DPP4 and the Adipose Tissue

In 2009, Kos and colleagues [99] demonstrated for the first time that DPP4 was expressed in both subcutaneous and omental fat depots [99], and one year later, a comprehensive proteomic profiling of the human adipocyte secretome identified DPP4 as one of the human-adipocyte-secreted proteins [100,101].

In the same period, DPP4 was identified as a novel adipokine [28]. In differentiated human adipocytes isolated from the AT, the DPP4 expression significantly increased during cell differentiation and was paralleled by a marked release of sDPP4; similarly, AT-derived macrophages released measurable amounts of DPP4, accounting for almost one third of adipocyte DPP4 secretion [28]. In adipocytes, treatment with DDP4 induced dose-dependent decrease of insulin-stimulated Akt phosphorylation, and consequent insulin resistance, demonstrating an autocrine effect of DPP4 in these cells [28].

As for clinical investigations, circulating sDPP4 levels were shown to be higher in obese subjects than age-matched lean individuals; studies on AT biopsies from the same patients revealed that AT DPP4 expression was influenced by both the total fat mass and the AT distribution [28]. Indeed, obese patients exhibited significantly higher DPP4 expression in both VAT and subcutaneous AT (SAT) in comparison to lean individuals, and DPP4 levels were higher in VAT than SAT. Finally, DPP4 expression levels positively correlated with adipocyte size and with the presence of metabolic syndrome [28].

Interestingly, AT explants from human VAT release more sDPP4 than those from SAT, leading to the hypothesis that the visceral compartment represents the main contributor to circulating sDPP4 levels in obese individuals [70].

Accordingly, other studies investigating the potential depot-dependent release of DPP4 from human SAT and VAT in a short-term incubation system confirmed that VAT is by far the major site of DPP4 release [102].

The role of DPP4 in adipocyte biology has been deeply investigated in human white preadipocytes and adipocytes, demonstrating that these cells express high amounts of DPP4 in all maturation stages, suggesting a possible contribution of DPP4 to the adipocyte differentiation process [103].

Mature adipocytes were identified as an important source of circulating DPP4. Progressively increased DPP4 protein expression and release were shown across more advanced adipocyte maturation stages [103]. The mechanism underlying increasing DPP4 release during maturation is not clear, as it has not been fully established which factors are able to trigger the DPP4 release.

Animal studies [104,105,106] investigating the effects of DPP4-Is treatment support the notion that DPP4 may play a functional role within the AT. In particular, the DPP4-I teneligliptin was shown to enhance brown AT function and to prevent obesity in mice on a high-fat diet [104]. A second study confirmed that teneligliptin prevented high-fat diet-induced obesity and increased energy expenditure in mice [105]. Treatment with des-fluoro-sitagliptin reduced white fat mass, favoring browning processes and energy expenditure in mice with diet-induced obesity [106].

Similarly, mice lacking the DPP4 gene were refractory to the development of obesity after a high-fat diet, in relation to a combination of reduced energy intake and concomitant increased energy expenditure [42]. Ablation of the DPP4 gene was also associated with improved metabolic control, in particular, improved insulin sensitivity, increased lipid oxidation, and reduced lipogenesis, in these models [42].

Other experimental data revealed that epigenetic modifications, rather than DPP4 gene polymorphisms, modulate tissue DPP4 expression [107,108]. Differential methylation levels within the DPP4 promoter were found to be associated with DPP4 mRNA expression in VAT from obese women with metabolic syndrome [108,109].

Several aspects of the DPP4 pathophysiology in relation to AT homeostasis are still unexplored. Whether greater DPP4 release from obese VAT reflects higher local DPP4 expression or depends on augmented and/or more efficient DPP4 cleavage has not been elucidated. Similarly, mechanisms of DPP4 release from cell membrane are still unknown, and the list of its substrates is not yet fully completed.

Finally, evidence from experimental studies shows that DPP4 is secreted from adipocyte and AT-associated macrophages in animal models and humans. DPP4 is released in great amounts from mature adipocytes, proportionally to the body fat mass extension, and mostly from visceral, rather than subcutaneous, fat depots.

DPP4 exerts autocrine effects on adipocytes, overall reducing insulin signaling in these cells, with unfavorable impact on metabolic outcomes under obesogenic conditions.

### 4.2. DPP4 and Sick Fat: Role of DPP4 in the Adipose Tissue Inflammation

Many studies have shown the contribution of DPP4 to AT inflammation, dysfunction, and remodeling.

Sell and collaborators demonstrated in obese subjects that VAT DPP4 expression and circulating sDPP4 levels were associated with increased adipocyte size, more pronounced VAT macrophages infiltration, and with circulating levels of other adipokines, such as interleukin 6 (IL-6) and leptin, and low adiponectin [70].

In a model of nutrient-induced visceral fat inflammation, based on high-sucrose and linoleic acid diet, which increased the mortality rate of diabetic mice, DPP4 inhibition enabled a nearly complete restoration of the mortality rate, together with a decrease of adipocytes size, CD8+ T cells, and M1 macrophages infiltration in VAT and reduced macrophage-related inflammatory genes expression [110,111].

Similar results were obtained when exploring the effect of the DPP4-I sitagliptin on AT inflammation in a diet-induced obesity model. In this investigation, in mice placed on a high-fat diet for twelve weeks, DPP4-I treatment significantly reduced VAT macrophage infiltration and adipocyte mRNA expression of inflammatory genes, including IL-6, tumor necrosis factor α (TNF-α), and interleukin 12 (IL-12) [112].

Similarly, in obese mice, treatment with the DPP4-I linagliptin reduced M1-polarized macrophage migration in VAT and induced M2-dominant shift of macrophage phenotype, thereby attenuating obesity-associated AT inflammation [113].

In addition to its direct effect on immune cells and inflammatory responses, DPP4 may also influence VAT stress, through its action on the extracellular matrix, with subsequent tissue inflammation and dysfunction. In vitro studies demonstrated that DPP4 impacts collagen molecules cleavage and degradation [22,23,114,115].

In obese mice and cell line of mouse adipocytes treated with the pro-fibrotic agent Transforming Growth Factor β1 (TGF-β1), the DPP4 inhibition prevented the increase of fibrosis markers and extracellular matrix deposition [116].

Figure 1 summarizes the effects of DPP4 in the AT in relation to inflammatory processes, VAT dysfunction, and insulin resistance.

### 4.3. DPP4 and Ectopic Fat Deposition

DPP4 was also investigated in relation to the deposition and accumulation of fat in ectopic sites, in particular in the liver, a condition known as nonalcoholic fatty liver disease (NAFLD) and, in its more severe form, steatohepatitis (NASH).

In DPP4-deficient mice fed with a high-fat diet, intrahepatic lipid accumulation was lower than in the wild-type mice, with significantly attenuated lipogenesis and increased fat oxidation, suggesting that mice lacking DPP4 were protected against high-fat-diet-induced hepatosteatosis [42].

Shirakawa and collaborators [110] explored the effects of des-fluoro-sitagliptin treatment on hepatic steatosis in animal models fed with a sucrose-rich diet and showed that DPP4 inhibition decreased the grade of steatosis at the liver histology, without impacting the overall liver weight [110].

Other studies [117] evaluated the effect of sitagliptin on the development of steatohepatitis in mice fed with a methionine/choline-deficient diet, a traditional diet model for NASH induction. DPP4 inhibition resulted in significantly lower hepatic fat accumulation and fatty acids uptake, along with reduced expression of very-low-density lipoprotein (VLDL) receptor and hepatic triglyceride content. Sitagliptin also effectively attenuated diet-induced hepatic inflammation, oxidative stress, and liver injury, as evidenced by reduced proinflammatory cytokine levels, endoplasmic reticulum stress markers, and decreased levels of fibrosis-associated proteins, such as fibronectin and α-smooth muscle actin (α-SMA), at the liver biopsy [117].

Some reports proposed that the hepatic, rather than the AT-derived, DPP4 influences the ectopic fat deposition and the development of insulin resistance [118,119], whereas a central role of adipocyte-derived DPP4 in the development of hepatic insulin resistance and diet-induced obesity was more recently shown [74].

AT-specific DPP4 knocked-out rodents displayed better glucose tolerance, higher insulin sensitivity, and lower hepatic fat content than wild types when fed with a high-fat diet; these mice had lower circulating sDPP4, confirming the major contribution of the AT as a source of serum sDPP4, along with the impact of AT DPP4 in NAFLD development in obesogenic conditions [74].

In addition to hepatosteatosis, the first data were very recently published on DPP4 levels in human epicardial adipose tissue (EAT) [120,121], which is known to accumulate in dysmetabolic conditions and is a marker of metabolic syndrome and increased cardiovascular risk [122]. EAT DPP4 levels correlated with VLDL cholesterol concentrations in SAT [120], and with worse renal function, and the presence of atrial fibrillation [121].

## 5. DPP4 in Metabolic Diseases

### 5.1. Obesity

As extensively discussed in the previous sections, it is substantially ascertained that circulating sDPP4 largely origins from the AT, thus attributing to this molecule the proper role of an adipokine [28]. Indeed, systemic sDPP4 levels depend on both the extension and localization of the fat mass; DPP4’s main source is the mature adipocyte sited in the visceral fat compartment, although it is also released from the SAT and from other cells within the AT, such as the resident macrophages [70,102,103]. Experimental evidence shows that, besides being a marker of AT inflammation and dysfunction, the DPP4, per se, elicits AT remodeling under stressful conditions, such as in the condition of chronic excessive caloric intake and obesity, and its inhibition reverts inflammation in animal models [110,111,112,113,114,115,116].

DPP4 also interferes with insulin signaling, reducing Akt phosphorylation, thus contributing to the development of insulin resistance [75]; for all these reasons, the detection of elevated sDPP4 levels in the presence of obesity is somehow expected [123,124,125,126,127]. Of interest, interventional studies reported beneficial effects of weight loss programs on circulating sDPP4 levels, in turn associated with improved metabolic outcomes [128,129].

Several investigations report increased circulating DPP4 concentration in obese individuals, mostly in the presence of metabolic syndrome [28,123,124]. In terms of body fat distribution, it has been observed that greater sDPP4 levels are associated with central fat deposition also in nonobese individuals, in whom it correlated with the presence of unfavorable metabolic risk profile [24].

The association between sDPP4, obesity, and the extension of VAT mass was also reported in adolescents [125], where DPP4 levels negatively correlated with basal and oral glucose tolerance test (OGTT)-stimulated GLP-1 secretion, confirming the relationship between higher sDPP4 levels and its increased enzymatic activity [125]. Despite differential incretin asset, in neither lean nor obese participants was sDPP4 associated with glucose tolerance [125].

The existence of a linear association between sDPP4 and BMI was also reported in individuals with T2DM, where short-term caloric-restriction-induced body weight loss was associated with decreased sDPP4 levels [126]; in turn, sDPP4 change correlated with the entity of visceral fat area reduction [127]. The same results were obtained when the weight loss was induced by intensive physical activity intervention [128]; indeed, in obese individuals, exercise decreased sDPP4 proportionally to the improvement of insulin sensitivity evaluated by euglycemic–hyperinsulinemic clamp, and DPP4 reduction correlated with greater basal fax oxidation [128].

Other investigations found that both bariatric surgery and nonsurgical weight loss interventions reduced sDPP4 in obese individuals [71]; however, sDPP4 changes were associated with improved serum transaminases and not with the weight loss entity or with improved markers of systemic inflammation [71]. These results may lead to different interpretations: on the one hand, they could reflect a direct role of DPP4 in hepatic damage in the presence of metabolic diseases, i.e., in the course of NAFLD/NASH, or at least a potential role of DPP4 as a biomarker of hepatic injury. On the other hand, since VAT inflammation is a major determinant of NAFLD [129,130,131,132], modifications within the AT environment, i.e., improved VAT inflammation and dysfunction after significant weight loss, may reduce the AT sDPP4 output, exerting beneficial effects on the intrahepatic fat accumulation (see also Section 5.3).

### 5.2. Type 2 Diabetes Mellitus (T2DM)

In addition to its role in facilitating body weight gain and the development of dysmetabolic phenotype in the condition of chronic caloric excess, DPP4 is also directly implicated in the control of insulin signaling and insulin secretion and release, as shown in a large number of experimental studies [70,71,72,73,74,75], and described in Section 2.2 and Section 2.4.

In clinical terms, it translates to the identification of a strong relationship between high circulating sDPP4 levels and the presence of T2DM, independent of obesity and other insulin-resistance-related disorders [25,126,133,134,135,136].

Ahmed and collaborators [126] found higher sDPP4 levels in subjects with T2DM compared to nondiabetic individuals; this association persisted after considering potential confounders such as the presence of metabolic syndrome and NAFLD.

In our previous reports [25,133], we demonstrated the existence of an association between higher sDPP4/DPP4 activity and the diagnosis of T2DM and metabolic syndrome [25]. Furthermore, we showed that higher circulating DPP4 activity represents a marker of early vascular dysfunction, expressed by impaired flow-mediated dilatation, in diabetic patients [133].

In our investigation, circulating sDPP4 was associated with higher fasting blood glucose, serum transaminases, and more severe insulin resistance, as estimated by the homeostasis model assessment for insulin resistance (HOMA-IR) in individuals with different degrees of obesity and glucose tolerance [25].

Plasma sDPP4 was shown to increase in nonobese individuals with T2DM compared to matched nondiabetic controls, and to correlate with the presence of impaired lipid profile and both subcutaneous and intra-abdominal fat deposition in metformin-treated adults [24].

Conversely, other studies reported increased sDPP4 levels in patients with T2DM, but the association with diabetes was mediated by the coexistence of obese phenotype [134].

Of note, circulating sDPP4 levels may predict clinical response to DPP-I treatment in T2DM [135]. In an exploratory pilot single-arm investigation, higher basal sDPP4 predicted poor response to sitagliptin therapy, in terms of 24-week glycosylated hemoglobin change, in T2DM patients previously treated with other oral antidiabetic agents [135].

Very recently, population-based studies and experimental models led to demonstrate that DPP4 genetic polymorphisms are involved in the control of post-OGTT excursions and subsequent C-peptide release responses through glucose sensing mechanisms that are abrogated in prediabetes [136].

Taken together, evidence to date points to the existence of a definite relationship between circulating sDPP4 levels and the presence of T2DM, which may be mechanistically explained by data showing the negative contribution of DPP4 on insulin sensitivity and β cell function.

Undoubtedly, part of the association between sDPP4 and T2DM is mediated by higher sDPP4 levels in the presence of obesity, which represents a common determinant of elevated DPP4 release and increased risk of insulin resistance and diabetes onset.

### 5.3. Nonalcoholic Fatty Liver Disease (NAFLD) and Steatohepatitis (NASH)

DPP4 has been largely investigated in many hepatic diseases, and initial evidence is available on the association between DPP4 and NAFLD.

Several data have shown the existence of a positive relationship between circulating DPP4 activity and the presence and severity of a number of chronic hepatic disorders, ranging from hepatitis C virus (HCV)-related hepatitis [137] and cirrhosis [138] to NASH [139] and NAFLD [140,141].

Indeed, the DPP4 activity in the bloodstream was proposed as a biomarker of liver impairment [142], although an association between the hepatic DPP4 expression and serum DPP4 was not detected [25].

Fewer data are available on sDPP4 and conditions associated with aberrant intra-hepatic fat accumulation, such as NAFLD and NASH. As summarized in the previous sections, elevated plasma sDPP4 levels depict conditions of altered metabolic homeostasis and insulin resistance, with sDPP4 concentrations that increase in presence of obesity, metabolic syndrome, and T2DM (see Section 5.1 and Section 5.2).

Similarly, initial evidence also associated greater sDPP4 with NAFLD in the presence of normal liver enzymes [25,133].

In our report [25], we investigated, for the first time, circulating sDPP4 and DPP4 activity contextually in relation to the presence of NAFLD, diagnosed with either liver biopsy or ultrasonography. Hence, besides confirming that NAFLD individuals display greater circulating DPP4 activity, we demonstrated the existence of a correlation between plasma sDPP4 and the steatosis grade, lobular inflammation, and ALT levels in severely obese individuals with biopsy-proven NAFLD [25]. The finding of an association between NAFLD and sDPP4 was confirmed in a second cohort of dysmetabolic individuals with or without obesity and T2DM [25].

More recently, sDPP4 was also investigated in relation to NAFLD-related liver fibrosis evaluated by transient elastography in T2DM individuals, demonstrating that sDPP4 was associated with the severity of liver fibrosis and was markedly elevated in diabetic patients with liver stiffness degree suggestive of cirrhosis [143].

NAFLD and NASH are hepatic manifestations of metabolic syndrome and are tightly associated with the presence of insulin resistance. Indeed, the existence of an association between sDPP4 and NAFLD/NASH may underlie a common pathogenic basis which involves increased fat accumulation and impaired insulin signaling, with suboptimal intrahepatic lipid oxidation and enhanced fat deposition.

However, more intriguing pathways have been described when investigating DPP4 in the context of gut–liver–adipose tissue axis impairment in metabolic disease. Experimental studies demonstrated that chronic caloric overload in obesity stimulates the production and secretion of sDPP4 by the hepatocytes, promoting VAT inflammation. Increased AT caveolin-1 release induces insulin resistance in the liver [119], suggesting that sDPP-4 may also be involved in crosstalk between the liver and adipose tissue.

Figure 2 summarizes the effects of DPP4 on adipose tissue, liver, and pancreatic islets which likely impair metabolic homeostasis in humans.

## 6. Conclusions

Starting from studies on the DPP4 enzymatic activities within the incretin system, many data emerged on the role of this molecule in controlling glucose/insulin homeostasis and immune inflammatory processes.

Growing evidence depicts a proper role for the DPP4 of an adipokine, whose circulating concentrations depend not only on the overall body fat mass, but also on its distribution and homeostasis. Indeed, if the adipose tissue has been, in general, defined as the main source of bioactive sDPP4, its secretion is strictly associated with fat accumulation within specific sites, such as the visceral compartment and the organ parenchyma of the liver in presence of NAFLD and NASH. 

Once DPP4 is secreted, it exerts autocrine and paracrine actions that converge on the modulation of insulin signaling and inflammatory processes. As an ultimate result, DPP4-dysregulated activity induces insulin resistance and chronic low-grade inflammation, particularly within the adipose tissue.

VAT remodeling is, in turn, one of the main actors in the development of metabolic disorders in overweight and obesity and is a determinant of insulin resistance and intrahepatic fat accumulation. Moreover, as DPP4 negatively impacts β cell function, the relationship between high sDPP4 levels and the presence of impaired glucose tolerance and T2DM is almost self-explanatory.

In animal models, AT DPP4 gene deletion is associated with the regression of VAT inflammation and improved glucose/insulin metabolism, though, so far, no specific cardio-metabolic protection in T2DM individuals chronically treated with DPP4-Is has been shown [144]. However, whether the enzymatic activity blocked by these agents involves the one exerted in the adipose tissue is still debated. Moreover, the circulating levels of sDPP4 were shown to be dissociated from the degree of systemic and AT inflammation [31], and this may explain why therapeutic agents modulating plasma DPP4 were not shown to protect from consequences of chronic low-grade inflammation, i.e., endothelial dysfunction and atherosclerosis, and from AT metabolic impairment in humans. In line with these considerations and with findings obtained in animal models, agents specifically modulating DPP4 at the AT level could potentially represent a valid therapeutic option for contrasting the development of metabolic disease in the presence of chronic caloric overload.

In conclusion, DPP4 is an adipokine with broad activities in multiple organs and systems and may determine unfavorable effects on metabolism. Despite the breadth of data on its action on the incretin system, many aspects of soluble DPP4 physiopathology and clinical effects of DPP4-tissue-targeted modulation still remain to be explored.

## Figures and Tables

**Figure 1 biomedicines-10-02306-f001:**
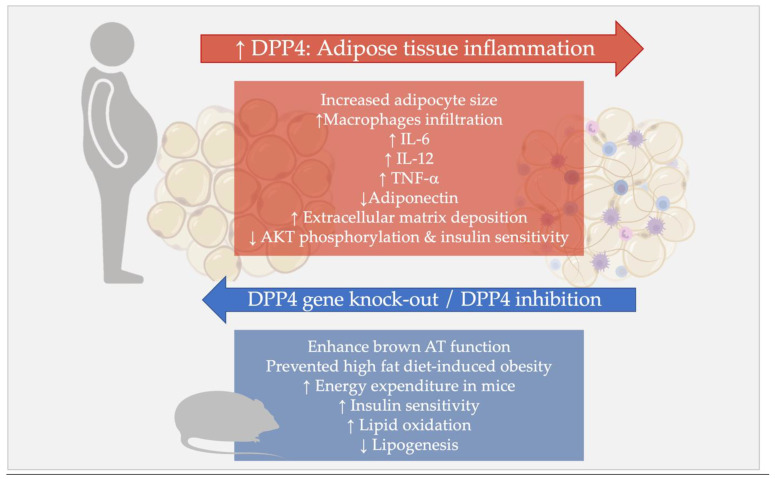
DPP4 in the adipose tissue. The red box shows the relationship between DPP4 and visceral adipose tissue inflammation, tissue remodeling, and dysfunction in humans. The blue box shows experimental data resulting from DPP4 ablation and pharmacological inhibition in rodents. Up arrows indicate an increase, down arrows a decrease. Abbreviations: DPP4: dipeptidyl peptidase 4, IL-6: interleukin 6, IL-12: interleukin 12, TNF-α: tumor necrosis factor α, AKT: protein kinase B; AT: adipose tissue. See Section 4.1 and Section 4.2 for detailed description and references.

**Figure 2 biomedicines-10-02306-f002:**
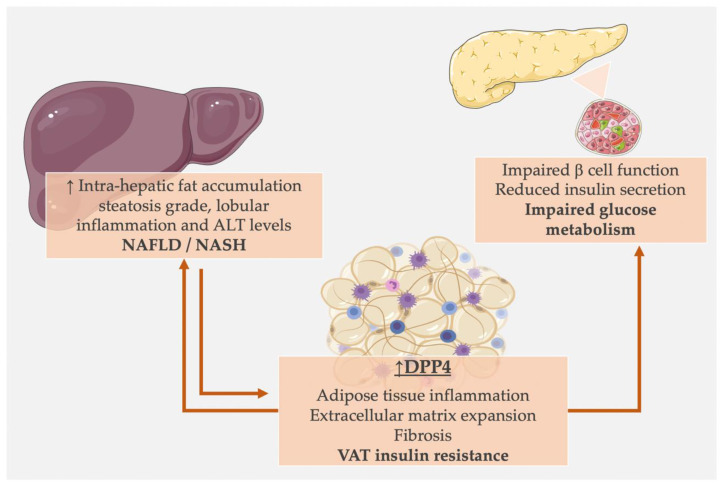
DPP4 in metabolic diseases: potential mechanisms linking high DPP4 to metabolic impairment. Arrows indicate the direction of the effects. Abbreviations: DPP4: dipeptidyl peptidase 4, ALT: alanine aminotransferase, NAFLD: nonalcoholic fatty liver disease, NASH: nonalcoholic steatohepatitis, VAT: visceral adipose tissue. See Section 4 and Section 5 for detailed descriptions and references.

## Data Availability

Not applicable.

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
