# Peer review of "Dipeptidyl Peptidase 4 (DPP4) as A Novel Adipokine: Role in Metabolism and Fat Homeostasis"

_biomedicines, 2022, doi:10.3390/biomedicines10092306_

Round 1

Reviewer 1 Report

The authors present an interesting review that describes the role of the adipokine DPP4 in the regulation of glucose, insulin and lipid metabolism; and its influence on different metabolic diseases such as obesity, T2DM, NAFLD, and NASH. This review was well written and concise, and the authors have previously published experimental articles about the role of DPP4 in different metabolic diseases.

Major comments:

-A huge number of citations are missing in several sentences and even paragraphs.

-Lines 138 to 141: How do you know the serum concentration of DPP4 peptides is low?. A robust reference is needed here. Please, include references when you affirm the unavailability of sensitive methods for the detection of DPP4.

-Line 199: Citation Zhang 2021 is not written in the appropriate style.

- Some acronyms and abbreviations are not defined the first time they appear in each of three sections. For example PI3K, Akt and ERK1/2 (Line 220), and VLDL (Line 520).

-In Figure 1, the description of the abbreviations are missing in the footnote.

-Please check, the appropriate style of every reference in the list of references. There are a lot of mistakes.

Minor technical questions/comments:

-English grammar, spelling, and connection between different sentences and paragraphs should be checked thoroughly in the manuscript.

-There are a lot of short paragraphs that could be synthesised into only one (Lines 125 to 136 as an example, but there are much more thoroughly the manuscript).

-When you indicate the fragments of the peptides between brackets, it is confusing with the references because it is the same format.

Author Response

The authors present an interesting review that describes the role of the adipokine DPP4 in the regulation of glucose, insulin and lipid metabolism; and its influence on different metabolic diseases such as obesity, T2DM, NAFLD, and NASH. This review was well written and concise, and the authors have previously published experimental articles about the role of DPP4 in different metabolic diseases.

Major comments:

-A huge number of citations are missing in several sentences and even paragraphs.

We thank the reviewer for his/her comment and have now revised the entire manuscript accordingly, adding missing citations.

-Lines 138 to 141: How do you know the serum concentration of DPP4 peptides is low? A robust reference is needed here. Please, include references when you affirm the unavailability of sensitive methods for the detection of DPP4.

We have now added new references of papers specifically addressing the challenges of DPP4 substrates measurements, in particular:  Baerts, L.; Waumans, Y.; Brandt, I.; Jungraithmayr, W.; Van der Veken, P.; Vanderheyden, M.; De Meester, I. Circulating stromal cell-derived factor 1alpha levels in heart failure: A matter of proper sampling. PLoS One 2015, 10, e0141408 (new reference 38) and Wang, W.; Choi, B.K.; Li, W.; Lao, Z.; Lee, A.Y.; Souza, S.C.; Yates, N.A.; Kowalski, T.; Pocai, A.; Cohen, L.H. Quantification of intact and truncated stromal cell-derived factor-1α in circulation by immunoaffinity enrichment and tandem mass spectrometry. J Am Soc Mass Spectrom 2014, 25, 614-25. doi: 10.1007/s13361-013-0822-7. Epub 2014 Feb 6. PMID: 24500701 (new reference 39).

Furthermore, we have now cited the important reference on the work by Elmansi AM et al (Elmansi, A.M.; Awad, M.E.; Eisa, N.H.; Kondrikov, D.; Hussein, K.A.; Aguilar-Pérez, A.; Herberg, S.; Peri-yasamy-Thandavan, S.; Fulzele, S.; Hamrick, M.W.; McGee-Lawrence, M.E.; Isales, C.M.; Volkman, B.F.; Hill, W.D. What doesn't kill you makes you stranger: Dipeptidyl peptidase-4 (CD26) proteolysis differentially modulates the activity of many peptide hormones and cytokines generating novel cryptic bioactive ligands. Pharmacol Ther 2019, 198, 90-108. doi: 10.1016/j.pharmthera.2019.02.005. Epub 2019 Feb 10. PMID: 30759373; PMCID: PMC7883480.) (new reference 40), which provides an extensive overview on challenges in sample collection, quantification techniques and data interpretation of DPP4 substrates, focusing on the cross-reactivity, short half-life and other issues limiting investigations in this field.

-Line 199: Citation Zhang 2021 is not written in the appropriate style.

This citation has been now reported in the text in the appropriate style (Page 5 line 208).

- Some acronyms and abbreviations are not defined the first time they appear in each of three sections. For example PI3K, Akt and ERK1/2 (Line 220), and VLDL (Line 520).

We have now defined all the acronyms and abbreviations the first time they appear in each section of the manuscript and in the figure’s footnotes, as indicated by the reviewer.

-In Figure 1, the description of the abbreviations are missing in the footnote.

The abbreviations list has been now added in the footnote of Figure 1 (Page 11, lines 553-554) and reference number in Figure 1 has been updated according to the revised reference list.

-Please check, the appropriate style of every reference in the list of references. There are a lot of mistakes.

We do apologize for the mistakes and have now checked and corrected the reference list.

Minor technical questions/comments:

English grammar, spelling, and connection between different sentences and paragraphs should be checked thoroughly in the manuscript. There are a lot of short paragraphs that could be synthesised into only one (Lines 125 to 136 as an example, but there are much more thoroughly the manuscript).

We thank the reviewer for these inputs and have now revised the entire manuscript, correcting all the grammar and spelling mistakes that we could find. The paragraphs indicated have been synthesised into one, according to this reviewer’s suggestion (Page 3, lines 133-139). Connections between sentences have been checked and short sentences have been synthesised to make the text more flowing and readable. All changes made are highlighted  in yellow in the new version of this manuscript.

 -When you indicate the fragments of the peptides between brackets, it is confusing with the references because it is the same format.

We apologize if indicating the fragments of the peptides between brackets was confusing and have now changed this format into superscripted round brackets.

Reviewer 2 Report

This review is well structured and well written.

Although I feel no significant concerns, the authors should refer to therapeutic considerations about lowering circulating levels of sDPP-4 or attenuating the expression of DPP-4 in the adipose tissues.   

Author Response

This review is well structured and well written.

Although I feel no significant concerns, the authors should refer to therapeutic considerations about lowering circulating levels of sDPP-4 or attenuating the expression of DPP-4 in the adipose tissues.   

We thank the reviewer for this comment and have added a new paragraph in the Conclusions (Section 6) commenting on potential therapeutic effects of DPP4 modulation on metabolism: “Moreover, the circulating levels of sDPP4 were shown to be dissociated from the degree of systemic and AT inflammation [31] and this may explain why therapeutic agents modulating plasma DPP4 did not show to protect from consequences of chronic low grade inflammation, i.e. endothelial dysfunction and atherosclerosis, and from AT metabolic impairment in humans. In line with these considerations and with findings obtained in animal models, agents specifically modulating DPP4 at the AT level, could potentially represent a valid therapeutic option for contrasting the development of metabolic disease in presence of chronic caloric overload” (Page 16, lines 794-801).

Round 2

Reviewer 1 Report

I believe the manuscript has been sufficiently improved to be published in Biomedicines.